# Transient expression of the neuropeptide galanin modulates peripheral-to-central connectivity in the somatosensory thalamus during whisker development in mice

Zsofia Hevesi[1,5], Joanne Bakker[2,5], Evgenii O. Tretiakov [1], Csaba Adori[2], Anika Raabgrund[1], Swapnali S. Barde[2], Martino Caramia[2], Thomas Krausgruber [3,4], Sabrina Ladstätter[3], Christoph Bock [3,4], Tomas Hökfelt [2,6] ✉ & Tibor Harkany [1,2,6] ✉

The significance of transient neuropeptide expression during postnatal brain development is unknown. Here, we show that galanin expression in the ventrobasal thalamus of infant mice coincides with whisker map development and modulates subcortical circuit wiring. Time-resolved neuroanatomy and single-nucleus RNA-seq identified complementary galanin (*Gal*) and galanin receptor 1 (*Galr1*) expression in the ventrobasal thalamus and the principal sensory nucleus of the trigeminal nerve (Pr5), respectively. Somatodendritic galanin release from the ventrobasal thalamus was time-locked to the first postnatal week, when $Gal_1R^+$ Pr5 afferents form glutamatergic ($Slc17a6^+$) synapses for the topographical whisker map to emerge. RNAi-mediated silencing of galanin expression disrupted glutamatergic synaptogenesis, which manifested as impaired whisker-dependent exploratory behaviors in infant mice, with behavioral abnormalities enduring into adulthood. Pharmacological probing of receptor selectivity in vivo corroborated that target recognition and synaptogenesis in the thalamus, at least in part, are reliant on agonist-induced $Gal_1R$ activation in inbound excitatory axons. Overall, we suggest a neuropeptide-dependent developmental mechanism to contribute to the topographical specification of a fundamental sensory neurocircuit in mice.

Neuropeptides are short amino acid sequences, and are classically viewed as neuroactive substances when released from dense core vesicles upon repetitive firing of nerve cells[1–3]. Once released, neuropeptides act on G protein-coupled receptors (GPCRs)[4] to modulate the activity of fast neurotransmitters in neuronal circuits in the adult brain[2], particularly for information processing during cognition and somatosensation[5,6]. For decades, the functional significance of the transient expression of some neuropeptides, e.g., proopiomelanocortin, cholecystokinin, neuropeptide Y, and somatostatin, remained unexplored during brain development, particularly at cellular foci that

[1]Department of Molecular Neurosciences, Center for Brain Research, Medical University of Vienna, Vienna, Austria. [2]Department of Neuroscience, Biomedicum 7D, Karolinska Institutet, Solna, Sweden. [3]CeMM Research Center for Molecular Medicine of the Austrian Academy of Sciences, Vienna, Austria. [4]Institute of Artificial Intelligence, Center for Medical Data Science, Medical University of Vienna, Vienna, Austria. [5]These authors contributed equally: Zsofia Hevesi, Joanne Bakker. [6]These authors jointly supervised this work: Tomas Hökfelt, Tibor Harkany. ✉e-mail: Tomas.Hokfelt@ki.se; Tibor.Harkany@meduniwien.ac.at; Tibor.Harkany@ki.se

are otherwise devoid of the expression of the same or any other neuropeptide in adulthood[7]. While neuropeptides have earlier been implicated in neural progenitor proliferation and the fate progression of γ-aminobutyric acidergic (GABAergic) neurons in both the cerebral cortex and hypothalamus[7–10], linking a specific neuropeptide to the formation and/or maturation of a defined neurocircuit in either the pre- or postnatal nervous system has so far eluded investigators.

In the early 1990s, galanin, a 29-amino acid-long neuropeptide[11], and its GPCRs (*Galr1*/Gal₁R, *Galr2*/Gal₂R, and *Galr3*/Gal₃R) were suggested by hybridization to undergo expressional waves in the neonatal brain[8,12,13]. A neurotrophin-like role for galanin in determining neuronal fate progression, survival, migration[14], and differentiation[15]—particularly neuritogenesis[16]—was suggested by the reduced numbers of basal forebrain cholinergic neurons in *Gal*[/-] mice. At the single cell level, galanin acting at Gal₁Rs in the growth cones of cholinergic neurons was shown to induce repulsive chemotropic steering decisions[17], at least in vitro. The hypothesis that galanin could affect neurocircuit maturation is also supported by the coincidence of transiently high galanin levels in the juvenile cerebellum (~postnatal day(P) 10) and the postnatal development of cerebellar white matter tracts, and a positive effect on local synaptogenesis[18]. Anatomical support for the concept that galanin could affect synaptogenesis, and the plasticity of newborn synapses, is from knock-in mice carrying *Galr1* or *Galr2* chimeras in which these receptors predominantly accumulate in putative dendritic spines, as well as presynapses in both brain and spinal cord[19]. Nevertheless, a gap of knowledge remains in understanding if manipulating the time-locked expression of neuropeptides, particularly galanin, manifests as altered behaviors later in life.

Here, we combined time-resolved neuroanatomy, mouse genetics, RNAi-mediated gene silencing, and receptor pharmacology in vivo to reveal that the somatodendritic release of galanin in the infant ventrobasal thalamus (VB) primes the wiring of Gal₁R⁺ afferent axons originating in the principal sensory nucleus of the trigeminal nerve (Pr5), thus contributing to the long-lasting stability of subcortical relay within the whisker pathway in mice. Consequently, reduced galanin levels during the P5-P10 time-window disrupt whisker-dependent exploratory behaviors.

## Results

### Transient galanin expression in the infant thalamus

First, we used radioactive riboprobe in situ hybridization to show galanin mRNA expression in the infant thalamus at P7 (Fig. 1a). Next, we corroborated this finding by single-cell resolved in situ hybridization (Fig. 1b, c) to demarcate a cellular locus of galanin expression in the VB. According to our immunohistochemical analysis, galanin expression peaked during postnatal days (P)4-10, then tailed off and completely disappeared by P21 (SI Fig. 1a). We then generated a *Gal*-Cre::Ai14 reporter mouse line to genetically identify cells that had expressed galanin during any period of their lifetime[10]. *Cre* recombinase protein in tdTomato⁺ neurons could indeed be histochemically detected in the thalamus around P7, but not in adulthood (Fig. 1d, e), confirming the transient activity of the galanin promoter. Nevertheless, tdTomato⁺ thalamic neurons survived into adulthood and became differentiated (NeuN⁺; Fig. 1f, f₁) in both female and male mice. These data characterize a cellular locus of transient galanin expression in the infant thalamus of mice, the physiological role of which could be associated with neurocircuit assembly because it is time-locked to a brief temporal window (P4-P10), with galanin expression coinciding with the enrichment of the VB in vesicular glutamate transporter 2⁺ (VGLUT2⁺) glutamatergic afferents (SI Fig. 1b–d).

### Galanin release from the ventrobasal thalamus

The VB represents the primary relay in the whisker pathway[20], linking the primary sensory nucleus of the trigeminal nerve (Pr5) and the barrel (somatosensory) cortex through its topographically mapped corticopetal afferents[21] (Fig. 1g). Both the axons (SI Fig. 2a, b) and the dendrites (Fig. 1h, SI. Fig. 2c) of VB neurons, the latter being innervated by Pr5 afferents, were galanin-positive(⁺) at P4-P7 (*see* SI Fig. 2a–c for galanin localization along dendrites in vitro and in vivo). Considering that the whisker pathway matures in an activity-dependent fashion postnatally when pups gain mobility[22], we hypothesized that galanin, if released from the VB, could contribute to the synapse establishment, selection, or maintenance of inbound Pr5 axons.

We used volumetric immunolabeling-enabled three-dimensional imaging of solvent-cleared organs (iDISCO⁺)[23] during the postnatal expansion of the VB (Fig. 2a–a₃) to reveal its gradual enrichment in tdTomato⁺ neurons (P1: $193 \pm 39$, P4: $346 \pm 33$, P7: $361 \pm 26$, P10: $604 \pm 30$, P14: $749 \pm 1$ cells/mm³; $p < 0.05$ for all ages *vs.* P1; Fig. 2b). Histochemical detection of galanin (*see also* SI Fig. 1b, b₁) demonstrated a rapid increase in the number of galanin⁺ thalamic neurons by P4, their maintained numbers until P10, and a significant reduction by P21 (for density, Fig. 2c). We next performed the time-resolved quantitative mapping of VGLUT2⁺ inputs to the VB, which showed that the density of VGLUT2⁺ afferents was high until P7, which is then followed by circuit refinement and synapse pruning (Fig. 2d). The size of VGLUT2⁺ presynapses could be used as a morphometric surrogate of synapse maturation, individually reaching up to 10 μm² after P10 (SI Fig. 1d). These data suggest that galanin expression peaks in the infant VB during postnatal glutamatergic synaptogenesis and maturation.

Next, we asked if galanin release could be evoked from thalamic neurons during P4–P7. When exposing organotypic slices prepared from P3, P6, P7, and P20 thalami to KCl (50 mM, used to indiscriminately release all vesicular content[24]), or brain-derived neurotrophic factor (BDNF; 100 ng/ml)[24], we detected significantly increased galanin levels in the bath solution only from thalamic slices prepared on P6/P7 ($p < 0.05$, in quadruplicate; Fig. 2e). These data were recapitulated by primary thalamic neurons plated on P3, but not P0, and maintained for 4 days in vitro (Fig. 2e₁), with immunocytochemistry from P3 cultures verifying the presence of galanin along their dendrites[25] in vitro (SI Fig. 2a, b). Cumulatively, these data suggest that galanin can be released from the somatodendritic compartment of neurons[25] populating the VB during the period of whisker circuit wiring.

### Complementary *Gal*/*Galr1* expression in the infant VB and Pr5

We posited that Pr5 neurons, which constitute the major sensory input of the whisker pathway (Fig. 3a), could use galanin for their axons to innervate thalamic relay neurons if they expressed galanin receptor(s) (*Galr1*, *Galr2* or *Galr3*). Because galanin induces repulsive growth cone turning[17], we used PC12 cells stably transfected with Gal₁R-GFP chimeras to test if either receptor could induce filo- or lamellipodial reorganization. We found that Gal₁R-GFP was particularly rapidly trafficked to the perikarya of PC12 cells, along with filopodial retraction, upon galanin (100 nM) stimulation (SI Fig. 2d). Thus, VB-derived galanin could indeed impact the distribution of Pr5 afferents during assembly of the whisker pathway, if a complementarity of galanin/ Gal(₁)R expression existed in the first postnatal week in mice. Indeed, multi-color in situ hybridization to co-localize *Gal* and *Galr1* mRNAs suggested the mutually exclusive expression of the ligand (*Gal*) and its cognate receptor (*Galr1*) in the VB and Pr5, respectively (Fig. 3b, c).

We then sought to reinforce this hypothesis by performing singlenucleus RNA-seq in parallel on the VB and Pr5 at P7. At this developmental stage, the VB contained two subtypes (cluster 1 and 3) of *Slc17a6*⁺ (glutamatergic) neurons, one subtype (cluster 6) of *Gad1*/ *Gad2*⁺ (GABA) neurons, astroglia, oligodendrocytes, and vascular components ($n = 923$ nuclei isolated from $n = 3$ pups of mixed sex; Fig. 3d, SI Fig. 3a, a₁). In the cluster of *Slc17a6*⁺ neurons, which we recognize as cortically-projecting glutamate neurons, ~12.2% expressed *Gal* (Fig. 3d₁). Besides *Gal*, these *Slc17a6*⁺ neurons co-expressed the thalamocortical neuronal marker *Cck*, as well as molecular components underpinning barrel map formation (e.g., *Grin1*, *Adcy1*, *Prkaca*,

*Ache, Slc6a4*). mRNA transcripts for other neuropeptides/hormones were absent or barely present (e.g., *Npy*, *Sst*; Fig. 3d$_2$). Galanin receptors (*Galr1, Galr2* or *Galr3*) were minimally, if at all, present in thalamic neurons (Fig. 3d$_2$). In turn, we harvested $n = 731$ nuclei from the Pr5 by micro-excision at P7, of which ~98% expressed *Slc17a6*, thus qualifying, in total or in part, as centrally-projecting sensory neurons (Fig. 3e; SI Fig. 3b, b$_1$). Amongst these neurons, ~3% expressed significant levels of *Galr1* (Fig. 3e$_1$). Neither *Galr2* nor *Galr3* was detected. This neuronal cohort also harbored axon guidance molecules (Fig. 3e$_2$). Thus, we suggest that galanin (and perhaps other neuropeptides) could participate in guidance decisions of Pr5 axons given the complementarity of ligand-receptor expression patterns, and that Gal$_1$Rs on Pr5 neurons could modulate neuritogenesis and/or directional steering decisions.

### RNAi-mediated Gal gene silencing impairs whisker-dependent behaviors

Interfering with *Gal* expression could reduce the number or delay the temporal dynamics of synapse maturation in the VB, if galanin acts as a chemotropic signal. If so, disrupted circuit wiring could impair exploratory behaviors reliant on tactile stimuli, particularly before eye opening (~P12 in mouse). Here, we used a 3D-printed stereotaxic apparatus customized for infant mice[26] (SI Fig. 4a) to administer *Gal*-targeting RNAi into the VB on P5; noting that efficient (~51%) knock-down was achieved within a day[27] in both sexes (Supplementary Fig. 4b, b1). We tested mice in a miniaturized open-field on P10, which revealed that mice that had received *Gal*-targeting RNAi were significantly less motile than those injected with scrambled RNAi ($p < 0.05$, $n = 6$/group; Fig. 4a, a$_1$, d). No difference was recorded in spatial preference between the center *vs.* peripheral zones of the field (Fig. 4b, b$_1$). *Gal* siRNA significantly reduced the density of VGLUT2$^+$ boutons apposing galanin$^+$ VB neurons (in both female and male pups; Fig. 4c-c$_2$; SI Fig. 4c-e$_2$). Notably, a close positive correlation between exploratory behavior and VGLUT2$^+$ synapse density in the VB was found (Fig. 4e).

Thereafter, we removed the whiskers and re-tested the infants. This manipulation significantly reduced general motility, particularly

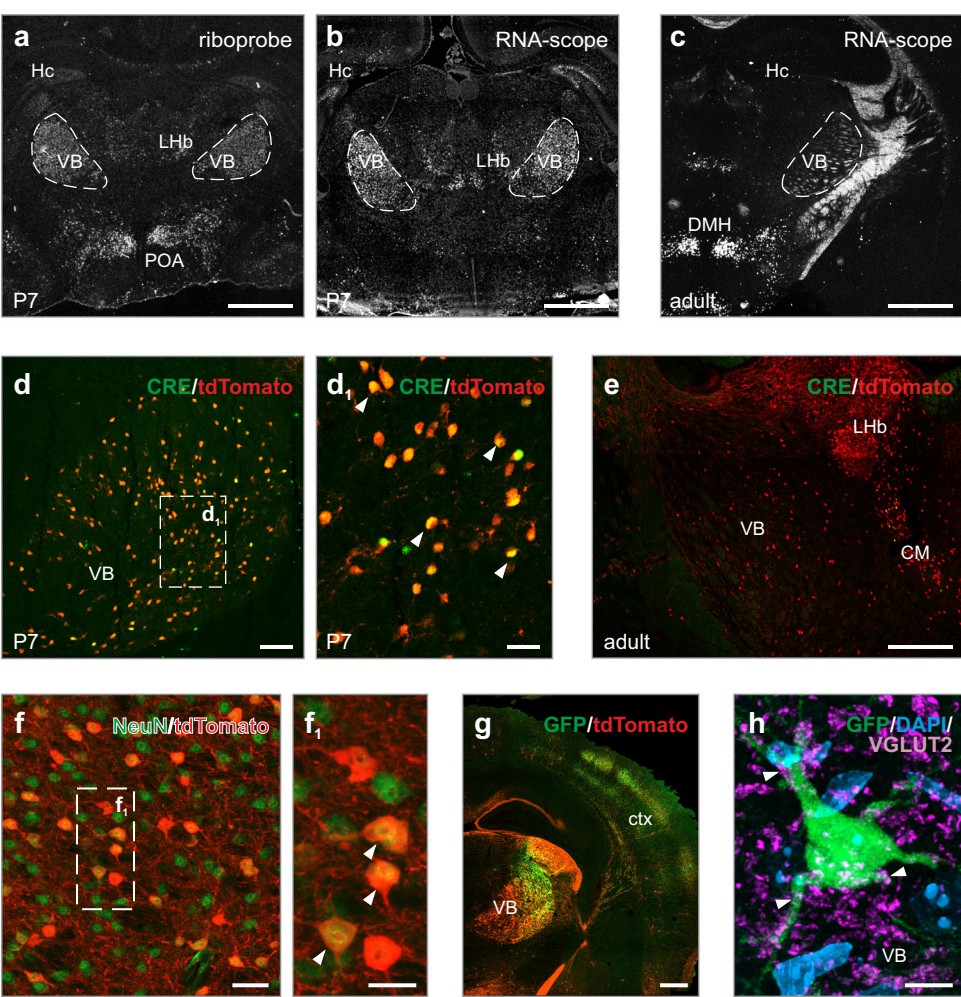

**Fig. 1 | Galanin expression is transient in the ventrobasal thalamus of infant mice.** Preprogalanin mRNA expression marked the ventrobasal thalamus (VB) on postnatal day 7 (P7), as shown by both a radioactive riboprobe **a** and chromogenic RNA scope in situ hybridization **b**. The panels illustrate different anteroposterior planes, but each is focused on the ventrobasal thalamus. **c** Preprogalanin expression was not detected in the adult brain. Lifetime tracing in *Gal*-Cre::Ai14 mice revealed tdTomato$^+$ cells in the VB. **d**, **d$_1$** *Cre* protein was only detected around P7 (*arrowheads*) in near-complete overlap with tdTomato, whereas *Cre* signal was absent in the equivalent area of adult mice **e**. **f**, **f$_1$** tdTomato$^+$ cells in *Gal*-Cre::Ai14 mice were neurons because of their co-labeling for NeuN (*arrowheads*). **g** tdTomato$^+$ neurons of the VB, and GFP/tdTomato co-labeling of cortical barrels (an adult mouse is shown). **h** A GFP$^+$ thalamic neuron is shown, which was apposed by vesicular glutamate transporter 2 (VGLUT2)$^+$ presynapses (*arrowheads*). *Abbreviations*: CM, centromedial nucleus of the thalamus; ctx, cortex; DAPI, 4′,6-diamidino-2-phenylindole; DMH, dorsomedial hypothalamic nucleus; Hc, hippocampus; LHb, lateral habenula; POA, preoptic area. Representative images are shown from $n > 3$ replicates processed independently. *Scale bars* = 500 μm **a–c, g**; 200 μm **e**; 50 μm **d$_1$, f**; 100 μm **d**; 25 μm **f$_1$**; 5 μm **h**.

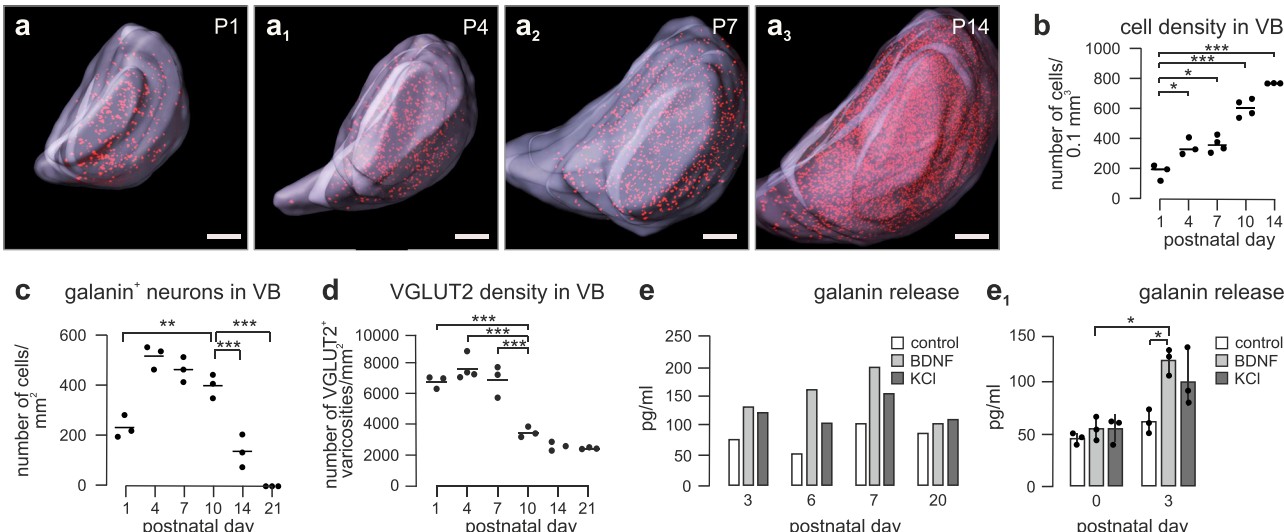

**Fig. 2 | Transient galanin expression and release in the ventrobasal thalamus.**
**a–a₃** Intact tissue imaging of the infant thalamus showed a progressive increase in cell density (**b**; $n = 3$ mice each for [P1, P4, and P14], $n = 4$ mice each for [P7, P10]), with immunohistochemistry showing transient galanin expression in neurons that peaked between P4-P10 and ceased by P21 (**c**; $n = 3$ mice for each age group). Glutamatergic synaptogenesis, measured as a density of VGLUT2⁺ varicosities (**d**; $n = 3$ mice each for [P1, 7, 10, 14, 21] and $n = 4$ [P4]), coincided with galanin expression, followed by synapse selection, and pruning from P14. Both brain-derived neurotrophic factor (BDNF, 100 ng/ml) and KCl (50 mM) evoked galanin release from organotypic slices prepared at P6/P7 (**e**; $n = 3$ biological replicates per age per condition) and primary neurons (**e₁**; $n = 3$ samples were pooled per age per condition) harvested at P3 and tested 4 days later. Note that galanin release was no longer possible at P20, consistent with the histochemical data **c**, **d**. Individual data points were plotted. Horizontal bars denote the means. Histochemical data (**b–d**) were statistically evaluated using one-way ANOVA followed by Tukey's multiple comparisons, and data of panel **e₁** were analyzed by two-tailed unpaired Student's $t$-test; $*p < 0.05$; $**p < 0.01$; $***p < 0.001$. The source data file is referred to for exact statistical output. *Scale bars* = 200 μm (**a₁–a₃**).

the exploration of the most stressful and unprotected central part of the arena, in controls after losing their vibrissae (*see* Fig. 4a₁,b₁ vs. Fig. 4f, f₁). In contrast, *Gal*-targeting RNAi-injected mice maintained their exploratory drive (that is, distance moved; Fig. 4a₁ vs. Fig. 4f) and even increased preference for the center (Fig. 4f₁, g). We interpreted these data as if whisker removal in intact animals occluded exploratory behaviors because of the acute loss of the primary sensory input. In contrast, reduced galanin expression in the VB limited the wiring of the whisker pathway and therefore rendered whisker removal irrelevant to the mice that received *Gal*-targeting RNAi. These behavioral data were supported by a negative correlation between exploratory behavior and VGLUT2⁺ synapse density in the VB (Fig. 4h). Cumulatively, our findings suggest that galanin could participate in establishing the wiring of central Pr5 afferents, thus shaping the peripheral arm of the whisker pathway.

Finally, we have left subgroups of mice injected with either *Gal*-targeting or scrambled RNAi on P5 to age until P45. Re-testing mixed-sex mice in a home cage-like setting[28] in the dark demonstrated significantly increased mobility in adulthood for those subjects that had received *Gal*-targeting RNAi during infancy (Fig. 4i). These data suggest the long-lasting resetting of somatosensory inputs through the whisker pathway, which are essential for exploration of a novel environment in the dark[28].

### Gal₁Rs modulate whisker-dependent behaviors

Galanin could act on one or more of its three GPCRs (Gal₁R, Gal₂R, or Gal₃R), which are expressed in the nervous system[29]. Even though our single-nucleus RNA-seq data implicated *Galr1* expression in Pr5 neurons in circuit maturation, we employed a pharmacological approach to confirm galanin receptor specificity. Therefore, either M617, a Gal1R-specific agonist[30,31], or M40, a non-selective galanin receptor antagonist[32,33], was microinjected acutely in the VB of P5 mice. Five days later, a significantly increased density of VGLUT2⁺ synapses was found in the VB of mice exposed to M617 (Fig. 5a–b₁). In contrast, M40

did not affect the density of glutamatergic synapses yet induced a marked increase in galanin levels (Fig. 5a). Both M617 and M40 increased exploratory behavior in infant mice of both sexes, with differential drug effects emerging upon whisker removal (Fig. 5c). We interpreted these findings as M617-induced Gal₁R gain-of-function, with a higher number of excitatory synapses inducing behavioral output, and even bypassing whisker removal, which is compatible with M617 inducing both neuroanatomical and behavioral phenotypes opposing the effects of RNAi-mediated *Gal* gene silencing. In contrast, the behavioral effect of M40 likely represents acute Gal₁R desensitization. Neither drug evoked long-lasting behavioral changes (Fig. 5d), which might be due to their significantly shorter half-life relative to that of RNAi-mediated gene silencing. Overall, these data support a role for Gal₁Rs in the formation and/or stability of Pr5 → VB afferent inputs.

## Discussion

Our study defines the role of the neuropeptide, galanin, in postnatal brain development. The fact that galanin modifies circuit assembly and wiring is not entirely unexpected if we consider that galanin, alike other signaling molecules that act via G protein-coupled receptors (e.g., endocannabinoids[34,35], Slits[36], circulating hormones[37]), could modulate directional steering decisions during axonal growth, and that galanin can be trophic to peripheral neurons[38,39].

Nevertheless, our results substantially expand the field of neuropeptide biology by highlighting that transient expression foci could specify neurocircuits, the impairment of which can manifest as altered behaviors. Considering that the expression of many neuropeptides, including galanin, is inducible[40–42], our study heralds an avenue of future research to address how changes in the spatial pattern, temporal distribution, and levels of neuropeptides in the infant/juvenile periods could contribute to the life-long determination of neurocircuit architecture, and if their alterations could be relevant to the pathobiology of congenital neuropsychiatric disorders.

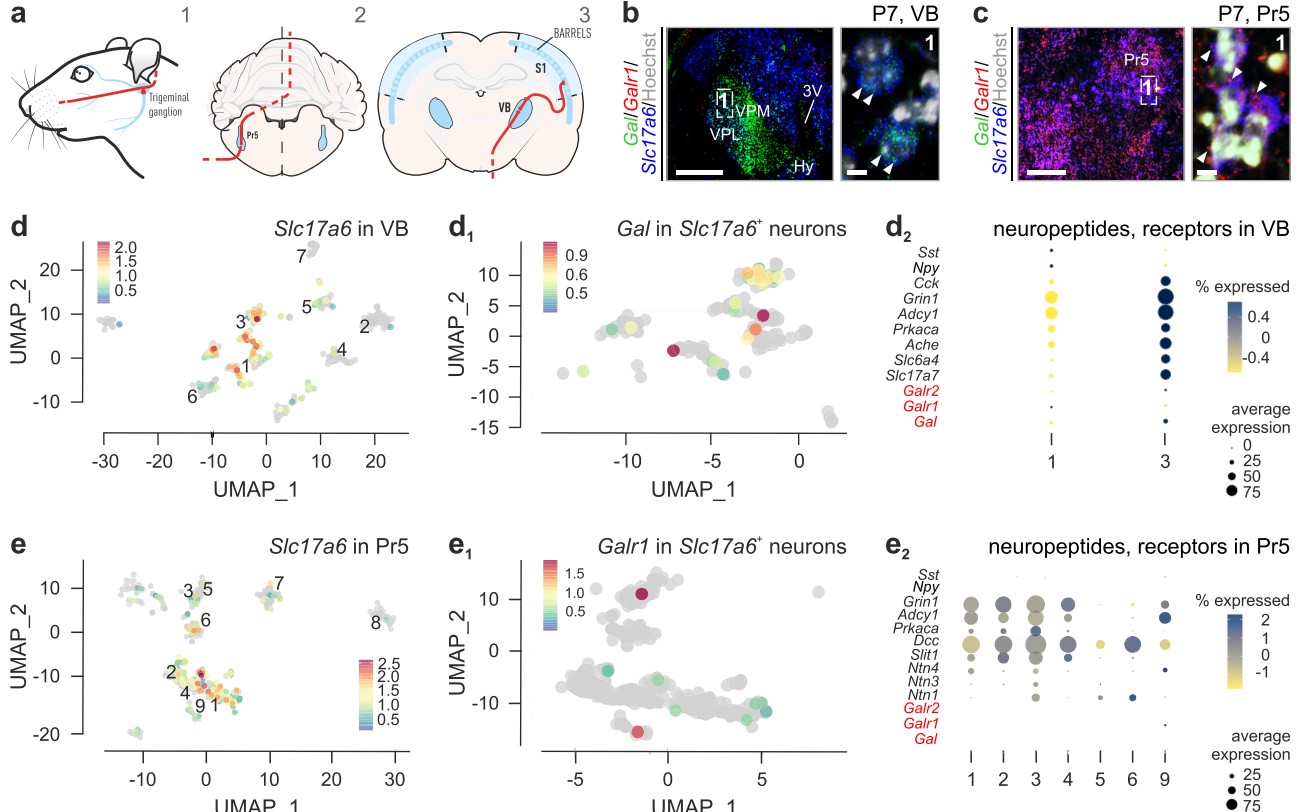

**Fig. 3 | Transient thalamic galanin expression complements *Galr1* expression in glutamatergic neurons of the principal sensory trigeminal nucleus (Pr5).**
**a** Schema of the somatosensory whisker pathway with its neuronal relay sites in the Pr5 and VB highlighted. **b** In situ hybridization revealed promiscuous *Gal* expression in the ventrobasal thalamus (VB) on postnatal day (P)7. Note that *Galr1* expression could not be detected in *Slc17a6*+ glutamatergic neurons of the VB (*arrowheads* in '1'). Conversely, the Pr5 contained *Slc17a6*+ neurons that harbored *Galr1* but not *Gal* expression (arrowheads, **c**). Open dashed rectangles in

**b**, **c** denote the positions of the insets ('1'). **d** Single-nucleus RNA-seq from the VB identified *Gal*+ neurons as glutamatergic (*Slc17a6*+; **d₁**) at P7, which also expressed many chemotropic guidance cues **d₂**. **e** Single-nucleus RNA-seq from the Pr5 revealed that *Slc17a6*+ sensory trigeminal projection neurons expressed *Galr1* at P7 **e₁**, offering receptor complementation for thalamic galanin expression, like other receptors for chemotropic guidance cues of the thalamus **e₂**. Panels **b**, **c** are representative of three independent experiments in *n* = 3 mice. *Scale bars* = 400 μm **b**, **c**; 7 μm (insets).

## Methods

### Animals
Experiments on live animals conformed to the 2010/63/EU directive and were approved by regional ethical authorities, and regulated by local laws (Stockholm Norra Djuretiska Nämnd: N631/19; Austrian Ministry of Science and Research: 66.009/0277-WF/V/3b/2017). Mice were maintained under standard conditions of husbandry, with a 12 h/12 h light cycle and 55% humidity. Care was taken to minimize the number and the suffering of the mice used. Animals of both sexes were used in all experiments, including wild-type C57BL/6 N and *Gal*-Cre mice (line KI87; BAC design, from GENSAT)[43]. The latter strain was crossed with heterozygous Ai14 reporter mice (B6.Cg-Gt(ROSA) 26Sor^tm14(CAG-tdTomato)Hze^/J), having a *loxP*-flanked STOP cassette preventing transcription of a CAG promoter-driven tdTomato, unless excised by *Cre* recombinase. Ensuing recombinant progeny were referred to as *Gal*-Cre::Ai14 mice throughout. Transgenic mice were backcrossed for multiple generations onto the C57BL/6 N background. *Gal*-Cre mice were bred and crossed heterozygously.

### Radioactive in situ hybridization using riboprobes
Prepro-galanin mRNA expression in mouse thalamus was first analyzed by radioactive in situ hybridization using riboprobes[41]. A 402-bp fragment of the mouse prepro-galanin cDNA (gene ID: NM_010253, corresponding to nucleotides 221-662) was generated by PCR from mouse total brain cDNA using GATGCCTGCAAAGGAGAAGAGAG *forward* and AGAGAACAGACGATTGGCTTGAG *reverse* primers. PCR

products were tested on 1% agarose gels. Amplicons were subcloned into a PCR1II-TOPO vector (Life Technologies). Probe sequences were confirmed by restriction digestion with *Xba1* (sense) and *BamH1* (antisense), followed by sequencing (KIGene).

Early postnatal (P7, *n* = 2) and adult mice (*n* = 2) were deeply anesthetized with isoflurane (5%, flow rate: 1 L/min), decapitated, after which whole brains were rapidly dissected out, and frozen on dry ice. Coronal sections of the thalamus were cut on a cryostat microtome (20 μm) and stored at −80 °C. Subsequently, the sections were sequentially immersed in ice-cold 4% paraformaldehyde (PFA) in 0.1 M phosphate-buffered saline (PBS; pH 7.4) for 10 min; 0.1 M PBS for 5 min; diethyl pyrocarbonate (DEPC)-treated water for 5 min; 0.1 M HCl for 5 min; 0.1 M PBS for 2 × 3 min; 0.25% acetic anhydride in 0.1 M triethanolamine for 20 min; and in 0.1 M PBS for another 2 × 3 min. Sections were then dehydrated in a graded series of ethanol (70–80–99.5%, for 2 min each), air-dried, and stored at −20 °C until further processing. For hybridization, sections were air-dried for maximally 1 h and incubated in a pre-hybridization cocktail [50% (vol/vol) deionized formamide (pH 5.0), 50 mM Tris HCl (pH 7.6), 25 mM EDTA (pH 8.0), 20 mM NaCl, 0.25 mg/mL yeast tRNA, and 2.5× Denhardt's solution] for 4−6 h at 55 °C, followed by hybridization in a humidified chamber overnight at the same temperature. Radiolabeled probes were prepared by in vitro transcription using the MAXIScript Sp6/T7 kit (Applied Biosystems) and [α³⁵S]UTP (NEG039H, Perkin Elmer). T7 and Sp6 polymerase were used to generate anti-sense and sense riboprobes, respectively. NucAway spin columns (Ambion) were used to separate the labeled probes from unincorporated

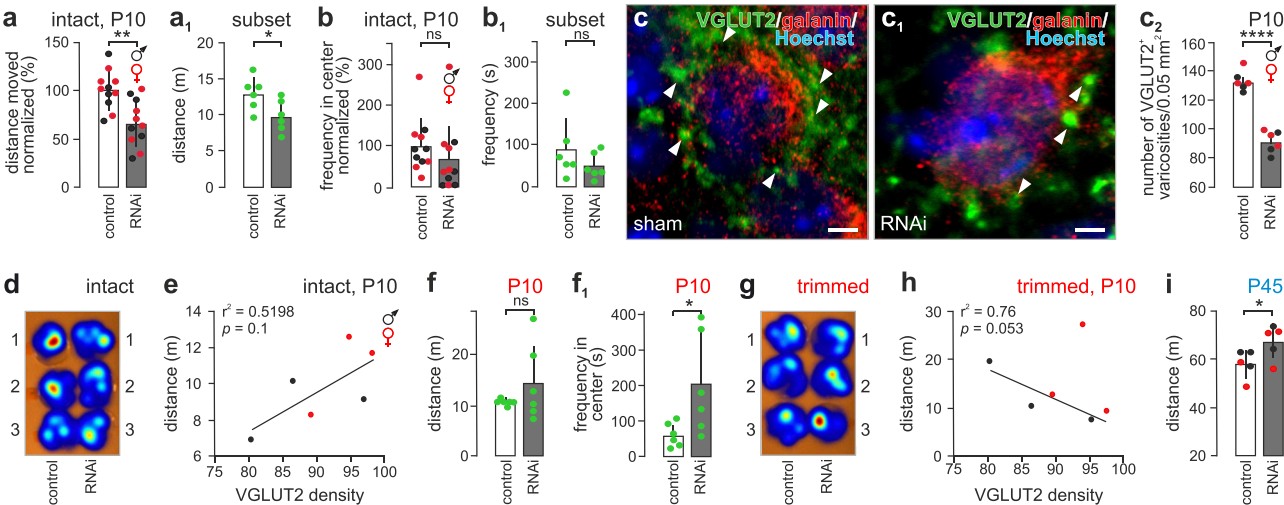

**Fig. 4 | RNAi-mediated galanin silencing impairs whisker-dependent exploration in infant mice. a–b₁** RNAi-treated mice were significantly less mobile, as compared to sham-operated controls (**a**, mixed for sex; $n = 11$ control vs. $n = 11$ RNAi-treated mice; $p = 0.002$), even though their center-to-periphery preference remained unchanged (**b**; $n = 11$ control vs. $n = 11$ RNAi-treated mice; $p = 0.314$). Panels **a₁**, **b₁** show randomized subsets of mice ($n = 6$ mice/group; $p = 0.032$ **a₁**; $p = 0.275$ **b₁**), which were later subjected to whisker plucking **f–h. c–c₂** The density of VGLUT2⁺ varicosities in the VB (*arrowheads*, **c**, **c₁**) was significantly reduced in infant mice injected with a *Gal*-targeting RNAi construct on P5 ($n = 6$ mice) and tested 5 days later, when compared to controls ($n = 6$ mice; $p < 0.0001$). **d** Heat maps illustrating the cumulative ambulation for control (*left*) and RNAi-injected mice (*right*; $n = 3$/group). **e** Positive correlation between the number of VGLUT2⁺ varicosities *vs.* behavioral responses to *Gal* RNAi ($n = 6$ mice). (**f**, **f₁**) Whisker

removal increased anxiety-like behavior in sham-operated mice ($n = 6$), as shown by the reduced time these animals have spent in the center of the arena. In contrast, RNAi-injected mice ($n = 6$) had significantly increased mobility ($p = 0.042$; *see data also against* **b**). **g** Heat maps of cumulative mobility after whisker removal, with $n = 3$ mice shown/group. **h** Negative correlation between the number of VGLUT2⁺ varicosities *vs.* behavioral responses to *Gal* RNAi after whisker removal ($n = 6$). **i** Long-lasting effect of *Gal* RNAi in the VB on animal mobility, tested on P45 ($n = 5$ controls *vs.* $n = 6$ RNAi-treated mice; $p = 0.039$). *Abbreviation*: ns, non-significant. Red and black data points denote female and male subjects, respectively. Data in bar graphs were expressed as means ± s.e.m. The effect of RNAi infusion was statistically evaluated using two-tailed unpaired Student's *t*-test throughout. *$p < 0.05$; **$p = 0.003$; ****$p < 0.0001$. *Scale bars* = 2 μm **c–c₁**.

nucleotides. Labeled probes were diluted to a concentration of $2.4–4.8 \times 10^6$ cpm/200 μL in a hybridization cocktail [50% (vol/vol) deionized formamide (pH 5.0), 0.3 M NaCl, 20 mM DTT, 0.5 mg/mL yeast tRNA, 0.1 mg/mL poly-A-RNA, 10% (vol/vol) dextran sulfate, and 1× Denhardt's solution]. Following hybridization, sections were washed sequentially in 1× SSC at 55 °C (2 × 30 min); 50% formamide/0.5× SSC at 55 °C (1 h); 1× SSC at 55 °C (15 min); RNase A buffer at 37 °C (1 h); 1× SSC at 55 °C (2 × 15 min); dehydrated in a graded series of alcohol (2 min each), and air-dried. Slides were first exposed to KODAK BioMax MR Film for 7–9 days, then dipped in KODAK NTB emulsion (Kodak; diluted 1:1 with water), exposed for another 4–9 weeks, developed in Kodak D19 developer, fixed in Kodak Unifix, and mounted in 90% glycerol/10% PBS (vol/vol). Dark-field images were captured on a Nikon microscope equipped with a Coolpix 5000 digital camera (Nikon). Brightness and contrast were manually enhanced. Multi-panel figures were assembled in CorelDraw 2022 v. 24 (Corel Corp.).

## RNA scope and HCR 3.0 fluorescence in situ hybridization
P7 ($n = 3$ [RNA scope]; $n = 4$ [HCR 3.0] and adult mice ($n = 2$) were deeply anesthetized with isoflurane (5%, flow rate: 1 L/min), decapitated, their brains rapidly dissected out, and frozen on dry ice. Serial coronal sections of 14-16 μm were stored under water-free conditions at −80 °C. Firstly, mRNA expression of prepro-galanin (#400961, mm*Gal*) was assessed using the RNA scope 2.5 HD RED kit (Advanced Cell Diagnostics), with slight modifications of the manufacturer's instructions (particularly, sections were fixed in 4% PFA-fixative at 4 °C for 1 h; treated with protease IV (22–24 °C) for 30 min before hybridization; and stained with Fast RED for 20 min)[44]. For HCR 3.0, hybridization probes against *Slc17a6*, *Gal*, and *Galr1* were used in both the Pr5 and VB[10].

## Immunohistochemistry
Adult mice were deeply anaesthetized with Na-pentobarbital, and transcardially perfused first with 20 ml saline (37 °C), followed by

20 ml warm (37 °C) fixative containing 4% PFA and 0.4% picric acid in 0.16 M phosphate buffer (pH 7.35). Perfusion was then continued with 50 ml of the same but ice-cold (4 °C) solution[45]. For infant mice (up to P14), anesthesia was induced by isoflurane (5% at 1 L/min flow rate), followed by perfusion with 15–20 ml of the above fixative at 22–24 °C. Brains were post-fixed for 2 h in the same fixative, and cryoprotected in 30% sucrose in 0.1 M phosphate buffer (PB) for at least 2 days. Coronal sections (20 or 40 μm) were then cut on a cryostat microtome, air-dried onto Superfrost⁺ glass slides (Thermo Fischer), and stored at −20 °C until processing. Sections were incubated in mixtures of primary antibodies (SI Table 1) for 2–3 days at 4 °C, followed by exposure to either directly-labeled secondary antibodies (Jackson ImmunoResearch, SI Table 2) or tyramide signal amplification (TSA Plus, Akoya Bioscience). In pre-absorption experiments, full-length galanin (Phoenix Pharmaceuticals) was diluted at $10^{-5}–10^{-7}$ M, and pre-absorbed with an anti-galanin antibody (1:8,000) overnight before proceeding with the TSA protocol ($n = 1$/age group). Adjacent sections incubated with anti-galanin antibody confirmed labeling specificity. Endogenous expression of tdTomato was sufficiently strong for imaging without additional enhancement by immunohistochemistry.

## Imaging and quantitative analysis
Representative images of immunostained sections were captured on an LSM 880 confocal laser-scanning microscope (Zeiss) using a one airy unit pinhole. For high-resolution images, a Plan-Apochromat oil objective (40×/1.3 N.A.; 63×/1.4 N.A.) was used. Images were processed in ZEN 2022 (Zeiss). Galanin⁺ somata, VGLUT2⁺ presynapses, and DAPI⁺ nuclei were counted manually using the Cell Counter plug-in in ImageJ (1.49 v; NIH) in maximum projection images from 18-μm orthogonal image stacks. Three-to-four sections/mouse from $n = 3$ mice/age group (except for P21 ($n = 2$); or $n = 6$ mice/experimental group for RNAi studies) were used. The average value was taken after counting in three different areas of the VB per section. For the quantification of the

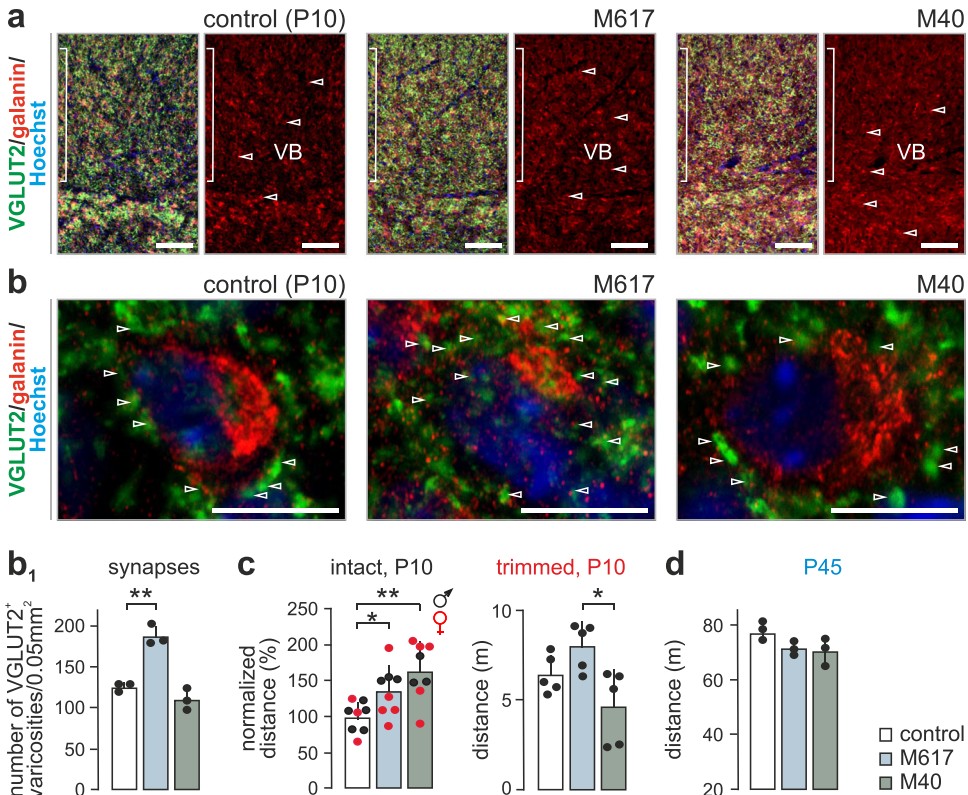

**Fig. 5 | Pharmacological manipulation of galanin receptors in the infant thalamus affects whisker-dependent behavioral outcomes. a** Distribution of galanin⁺ neurons and VGLUT2⁺ afferents in the ventrobasal thalamus (VB) on postnatal day (P)10. M617, a Gal₁R-selective agonist, and M40, a non-selective antagonist, were applied locally on P5. Open *arrowheads* point to galanin⁺ peri-karya. Range indicators to the left mark the altered density of VGLUT2⁺ afferents. Photomicrographs are representative of each experimental group in **b₁**. **b** Gal₁R modulation affected the density of glutamatergic (VGLUT2⁺) synapses in the VB. Note the opposing effects of M617 and M40. **b₁** Morphometric data were collected from *n* = 3 mice/group, and statistically assessed using two-tailed unpaired

Student's *t*-test (treatment *vs.* control). M617 significantly increased synapse density (**$p$ = 0.0014). **c** Gal₁R manipulation significantly increased exploratory behavior in infant mice on P10 (*n* = 8 mice/group; male (black) and female (red) data were disaggregated as shown). Data were analyzed by Student's *t*-test (two-tailed, unpaired): *$p$ = 0.023 (control vs. M617), **$p$ = 0.001 (control *vs.* M40). However, M617 and M40 had opposite effects after whisker removal, recapitulating RNAi-induced outcomes (*n* = 5 mice/group; *$p$ = 0.015 (M617 vs. M40); two-tailed unpaired Student's *t*-test. **d** The effect of acute pharmacological manipulation of Gal₁Rs in the VB was transient, as assessed on P45 (*n* = 3 mice/group). Data in bar graphs were expressed as means ± sem throughout. *Scale bars* = 100 μm **a**, 10 μm **b**.

density of VGLUT2⁺ presynapses during the early postnatal period, 2-μm orthogonal image stacks were captured at 40× magnification at three non-overlapping subregions of VB. Two sections/mouse from *n* = 3 mouse/age group were used, except at P4 (*n* = 4) and P21 (*n* = 2). Orthogonal image stacks were analyzed in Imaris (8.4.2TM; Bitplane). The total number of VLGUT2⁺ puncta was obtained in 0.05 mm² surface areas.

## iDISCO⁺, image acquisition, and data processing

The brains of *Gal*-Cre::Ai14 mice (*n* = 2 per age, processed and analyzed as 4 independent brain hemispheres) were collected after immersion-fixation in PFA fixative (as above; P1, 20 h) or perfusion with and post-fixation in the same solution (P4: 16 h; P7: 4–6 h; P10: 4–6 h; P14: 2 h). These terminal procedures were carried out under deep anesthesia with isoflurane (5%, flow rate: 1 L/min). Brains were kept in 0.01 M PBS at 4 °C until processing. iDISCO⁺ volumetric immunostaining and tissue clearing were performed as described[23]. Briefly, intact hemispheres (P1, P4, P7, and P10) or half forebrain blocks (P14) were washed in 0.01 M PBS (3 × 5 ml) in Eppendorf tubes, and then dehydrated in an ascending series of methanol/water (20%, 40%, 60%, 80%, and 2 × 100% methanol) for 1 h each. Samples were bleached with 5% $H_2O_2$ in 100% methanol at 40 °C overnight, rehydrated, incubated in permeabilization solution for 2 days, and then in blocking solution for another 2 days, both at 37 °C (0.2% Triton-X100/20% DMSO/0.3 M glycine/0.02% NaN₃ in 0.01 M PBS and 0.2% Triton-X100/10% DMSO/6% normal

donkey serum/0.02% NaN₃ in 0.01 M PBS, respectively). Samples were then exposed to a primary antibody-containing solution (anti-RFP, 1:300, Rockland, #600-401-379) in 0.2% Tween-20/10 μg/ml heparin/5% DMSO/3% normal donkey serum/0.02% NaN₃ in 0.01 M PBS at 37 °C for 5 days. After extensive washing, tissue blocks were incubated in a secondary antibody solution (1:300; Alexa Fluor 647-conjugated goat anti-rabbit, Molecular Probes) in 0.2% Tween-20/10 μg/ml heparin/3% normal donkey serum/0.02% NaN₃ in 0.01 M PBS. Tissues were then dehydrated in an ascending series of methanol (20%, 40%, 60%, 80% and 2 × 100%) for 1 h each, incubated in 66% dichloromethane/33% methanol for 3 h, followed by 100% dichloromethane for 2 × 15 min. Subsequently, tissue blocks were moved to tubes filled with 100% dibenzyl ether, and stored in this solution for the long term. A light-sheet microscope (Ultramicroscope II, Lavision Biotec) and Imspector™ software were used to acquire the samples. The microscope was equipped with an sCMOS camera (Andor Neo) and a 2×/0.5 MVPLAPO 23 objective with a 6.5-mm working distance (spherical aberration corrected dipping cap). The samples were fixed in the sample holder, and thalamic regions were acquired coronally in multi-color 3D scanning mode (488 nm: tissue autofluorescence; 647 nm: RFP immunosignal). Microscope parameters were as follows: 2× objective, 4× zoom body and additional magnification of the dipping cap lens (altogether 9× effective magnification; 0.76 μm × 0.76 μm × 2 μm voxel size), 100% and 80% laser power for 647 nm and 488 nm, respectively, single-side illumination, 120 ms exposure time, max sheet

numerical aperture (0.156), dynamic horizontal focus process with 15 steps (contrast adaptive algorithm), 60% sheet width, 2 μm z-step thickness. Series of 16-bit uncompressed TIF images were then converted to IMS files, and the samples were reconstructed in Imaris (9.2.1TM; Bitplane). For quantification of the number of RFP$^+$ neurons within the VB in 3D, the thalamus was segmented based on autofluorescence (488 nm channel), using the 'manual surface creation' function in Imaris with 1.5-μm surface grain size. 'Surface volume' (μm$^3$) and 'sphericity' were determined. 3D segmentation by autofluorescence was also applied to the 647 nm signal, creating a masked channel. The RFP immunosignal was amplified and noise-filtered in Fiji-ImageJ/Imaris XT using contrast-limited adaptive histogram equalization (CLAHE), background subtraction, and minimum filtering. The number of RFP$^+$ neurons was determined in 3D by the spot-detection function of Imaris, where each RFP$^+$ cell was defined as a spot (parameters: estimated X/Y-diameter of a spot: 10 μm, z-diameter: 20 μm, no background subtraction, 'quality' filtering). The accuracy of the spot detection algorithm was carefully checked after each scan. 'Total number of dots' and 'average of spot-to-spot closest distance' were then determined.

### Quantitative PCR

Quantitative PCR reactions were performed on dissected thalami of siRNA-injected and control mice (n = 3/group/age) that had been anesthetized with isoflurane (5%, flow rate: 1 L/min). Total RNA was extracted by the RNA mini kit (Bio-Rad), and its concentration determined on a Nanodrop. RNA samples were reverse-transcribed using a high-capacity cDNA reverse transcription kit (Thermo Fisher). Reactions were performed after an initial denaturation step at 95 °C for 3 min followed by 35 cycles of 95 °C for 1 min denaturation, annealing and extension at 60 °C (1 min each), and a dissociation stage at 72 °C (2 min) on a CFX 96 apparatus (Bio-Rad). Primer pairs (SI Table 3) amplified short fragments of the mouse *Gal* and *Gapdh* genes.

### Primary neuronal cultures and organotypic slices

Primary neuronal cultures were prepared from either P0 or P3 C57BL6/JRj mouse brains (n = 5–7/litter; both sexes). Thalami were rapidly dissected, collected in bulk, rinsed 2× with ice-cold Hank's balanced salt solution (HBSS; Gibco), and then exposed to a papain dissociation solution (Worthington) at 37 °C for 45 min. After washing 2× with warm HBSS, tissues were homogenized in 3 ml DNase (Sigma) by trituration, followed by incubation at 37 °C for 10 min. Next, samples were pelleted (1000 *rpm*, 22–24 °C, 4 min) in Neurobasal medium (Thermo Fisher) also containing 1% B27 supplement (Gibco). The pellet was resuspended in 2 ml Neurobasal medium and filtered with a 40-μm cell strainer into a Falcon tube. Cells were plated at a density of 150,000-180,000 cells/well in poly-D-lysine-coated 24-well plates. On day 4 in vitro, cells were fixed with 4% PFA for immunohistochemistry. Alternatively, neurons were treated with BDNF (100 ng for 48 h, 4–6 days in vitro) or stimulated by 50 mM KCl for 30 min on DIV6. Thereafter, supernatants were collected for the ELISA-based detection of galanin. Samples were kept at −80 °C until processing.

Organotypic slices spanning the VB were prepared at P3, P6/P7 and P20 (C57BL6/JRj of both sexes) by rapidly dissecting whole brains, and placing them into ice-cold HBSS. After embedding the brains in 4% low-melt agarose (Sigma), 300-μm thick coronal slices were made on a vibratome (speed: 0.3, vibration: 1.25; Leica VT1200), and transferred onto Millicell inserts (2 slices of different origin/insert), which were then submerged in 1 ml 10% FBS-containing DMEM (Gibco) in 6-well plates. DMEM/FBS was replaced with Neurobasal medium 2 h later. On the next day, slices were treated with BDNF (100 ng/ml) for 2 days. Alternatively, on day 4 in vitro, slices were exposed to 50 mM KCl for 30 min. Supernatants were collected to determine extracellular galanin levels by ELISA. Samples were processed in batches and kept at −80 °C until processing.

### PC12 cells and over-expression of Gal$_1$Rs

Gal$_1$R-EGFP vectors were made by reverse transcribing the open reading frame of the rat *Galr1* gene, followed by PCR amplification using the primers as shown in SI Table 3. PCR products were ligated into a pEGFP-NI vector (Clontech) at its *HindIII/SacII* and *Hind/KpnI* restriction sites to generate *Galr1*-EGFP expression vectors. Rat pheochromocytoma cells (PC12) (1,000,000 cells/plate) were transfected with *Galr1-Egfp* (1 μg plasmid DNA) using Effectene Transfection Reagent (Qiagen), as described earlier[46]. Stably transfected cells were selected by growing them in the presence of geneticin (G418, Sigma) at a concentration of 800 μg/ml. PC12 cells were sub-cultured and used for 10–15 passages. For Gal$_1$R-EGFP internalization, cells were seeded on 13-mm glass coverslips (10–20,000 cells/well in 24-well plates) previously coated with poly-D-lysine (Sigma), and pharmacologically probed 72 h after plating. Galanin was superfused at a concentration of 100 nM. Cells were photographed using differential interference contrast (DIC) microscopy (DMLFSA, Leica).

### Galanin detection by ELISA

Supernatants were concentrated in Amicon Ultra-3K centrifuge inserts (3 kDa cut-off, Sigma; molecular weight for galanin is calculated at 3.157 kDa) through repeated centrifugation steps according to the manufacturer's recommendations (14,000 g, 4 °C, 30 min followed by 1000 g, 4 °C, 2 min). Protein concentrations of the samples were determined by the BCA method (Thermo Fisher), and set at 2 μg/μl throughout. Galanin levels were measured by an ultrasensitive mouse galanin kit (Abcam, #272209).

### Single-nucleus RNA-seq, and bioinformatics

Seven-day old mice (n = 3, C57Bl6/JRj of mixed sex) were anesthetized (isoflurane; 5%, flow rate: 1 L/min), perfused with ice-cold modified HEPES solution containing (in mM): 90 NaCl, 3 KCl, 1.2 HEPES, 20 Na-pyruvate, 3 Na-ascorbate, 5 NaHCO$_3$, 30 D-glucose, 25 MgSO$_4$, 2 CaCl$_2$, 1.5 L-glutathione), the brains dissected, and sliced at 400-μm thickness on a vibratome (settings: speed 0.3, vibration 1.4; Leica VT1200). The VB and Pr5 were simultaneously punched out under a stereomicroscope and collected in separate cryovials. The samples were snap-frozen using isopentane to minimize the formation of ice crystals and to preserve tissue integrity. Samples were kept at −80 °C until RNA extraction, single-nucleus library preparation, and RNA-sequencing.

For single-nucleus RNA-seq, nuclei were isolated from snap-frozen tissues by direct dissociation in cold Nuclei EZ Lysis buffer (Sigma; NUC101) using a pestle. Lysates were incubated on ice for 10 min, filtered through 70-μm strainer meshes, and centrifuged at 500 g, at 4 °C for 5 min. Pelleted nuclei were washed in 'nuclei wash and resuspension buffer' [1% BDA in 1× DPBS plus 0.5 U/μl Protector RNase-Inhibitor (Sigma: 3335402001)], followed by centrifugation at 500 g at 4 °C for 5 min. Pelleted nuclei were then washed 3× before taking them up in 'nuclei wash and resuspension buffer' supplemented with 7-amino-actinomycin D (7-AAD, BioLegend; #420404). After straining through a 40-μm mesh, single nuclei were sorted using a SONY SH800S cell sorter. Sorted nuclei were processed for single-nuclei RNA-seq using the Chromium Next GEM Single Cell 3′ Reagent Kit v3.1 (10× Genomics; #PN1000128) according to the manufacturer's instructions. Data preprocessing, quality control (including droplet selection, ambient RNA removal, doublet detection, and refined filtering), clustering, marker gene definitions, and the specific software packages used for data processing were made available at https://harkany-lab.github.io/Hevesi_2023/eda.html#Separate_analysis.

### Pre-processing

The Cell Ranger pipeline (v7.1.0)[47] was used to perform sample demultiplexing, barcode processing, and gene counting per nucleus. Briefly, samples were demultiplexed to produce a pair of FASTQ files per sample. Reads containing sequence information were aligned

using the optimized mouse genome reference (vmm10_optimized_v.1.0)[48] on the default Cell Ranger mm10 genome version 2020-A, which was cleared from gene overlaps, poorly annotated exons, 3′-UTRs and intergenic fragments. PCR duplicates were removed by selecting unique combinations of cell barcodes, unique molecular identifiers (UMIs), and gene identifiers, with the results forming a gene expression matrix.

### Droplet selection
The droplet selection method integral to Cell Ranger and operating under the *EmptyDrops* method[49,50] identified 923 and 731 nuclei in VB and Pr5, respectively.

### Ambient RNA removal
We applied *CellBender*[50], a neural network-based approach (docker://etretiakov/cellbender:v0.0.1). A false positive rate threshold was set at 0.01, with the neural network run to learn over 150 epochs with a total of 5000 droplets included (based on knee plots; see Cell Ranger Reports at GitHub).

### Doublet detection
Log probability was qualified as a doublet for every cell based on a priori knowledge of genotypes from each input sample, and called variant occurrence frequencies that allowed to cluster nuclei to either the source organism or be classified as a doublet (shub://wheaton5/souporcell).

### Further filtering
Information to annotate genes was added using the *gprofiler2* package (v0.2.1). Accordingly, we filtered cells based on their high content of mitochondrial, ribosomal, or hemoglobin-coding genes. Thresholds were set as 10%, 1%, and 0.5%. Additionally, pseudogenes and poorly annotated genes were deleted from the count matrix. Cells of low complexity were filtered out as $\log_{10}(\text{genes})/\log_{10}(\text{UMIs}) < 0.8$. Cells were assigned cell cycle scores using the *CellCycleScoring* function in the *Seurat* package (v4.3.0).

### Clustering
**Gene selection.** We used the selection method integral to the *Seurat* package (v4.3.0)[51,52], which uses a modern variance-stabilizing transformation statistical technic that utilizes scaling *Pearson* residuals[53]. Accordingly, we selected 3000 highly variable genes/dataset, and regressed out complexity and cell-cycle variability prior to the final scaling of filtered matrixes.

**Graph-based and multi-level clustering.** We performed *Leiden* algorithm graph-based clustering. PCA was performed using selected genes and the jackknife tested[54]. Principal components (we tested the significance of feature for randomly picked 1% of data over 1000 iterations; see *PCScore* function in the *functions.R* script of the code directory) were used to construct a shared nearest-neighbor graph using the overlap among the 15 nearest neighbors of each cell. Leiden modularity optimization[55] was performed to partition this graph with an array of resolution parameters, where 30 modularity events were sampled between 0.2 and 2.5. Clustering trees[56] were visualized by the *clustree* package (v0.5.0). The resolution of previously identified clusters was reconciled by the *mrtree* package (v0.0.0.9000)[57]. We calculated the adjusted multi-resolution Rand index chosen as the maximum value, if there had been no higher modularity within 0.05 AMRI difference (see *SelectResolution* in *function.R* file of the code directory).

**Marker genes.** Marker genes for each cell cluster were identified using the *logreg* test[58] implemented in the Seurat framework[52]. Genes were considered significant markers for a cluster if their FDR was <0.001.

Identities were assigned to each cluster by comparing the detected genes to previously published markers, as well as our own validation experiments, particularly,: 'GO:0007411' for axon guidance, 'GO:0007409' for axonogenesis, 'GO:0030426' for growth cone, 'GO:0021860' for pyramidal neuron development, 'GO:0097490' for sympathetic neuron projection extension, 'GO:1990138' for neuron projection extension, 'GO:0010976' for positive regulation of neuron projection development, 'GO:0010977' for negative regulation of neuron projection development, and 'GO:0021952' for central nervous system projection neuron axonogenesis.

**Stereotactic surgeries, virus injection, and RNAi-mediated galanin knock-down in vivo.** Wild-type (P5, $n = 46$) and *Gal*-Cre::Ai14 (P7, $n = 11$) mice pups were used for stereotactic injections performed on a custom-made 3D-printed stereotaxic adapter for infant mice[26]. Mice were anesthetized by inhalation using isoflurane (1.5%, 1 L/min airflow). The injection procedure was essentially identical to those in adult mice[44]. Body temperature was maintained with a warming pad, and carefully monitored throughout the surgical procedure. Instead of using a drill, the skull was pierced with a 27 G needle to remove a minimal skull fragment, thus allowing the insertion of a glass capillary syringe.

*Gal*-Cre::Ai14 pups were bilaterally injected with 100–200 nl ($8 \times 10^{12}$ particles/ml) of rAAV2-FLEX-Halo57-GFP (AV 5014, Virus Vector Core) at a speed of 100 nl/min (coordinates: anterior-posterior (AP): −1.4/−1.5 mm from bregma, mediolateral (ML): ±1.5 mm, dorsoventral (DV): 3.1-3.3 mm from the dura). Injection coordinates for both in vivo RNAi knock-down and pharmacological experiments were as follows: AP: −1.2 mm from bregma, ML: ±1.4 mm, DV: 3.0 mm from the dura. One hundred and fifty nl of Accell galanin RNAi solution or GFP-tagged scrambled control (Dharmacon) were injected. Alternatively, 75 nl of either M617 (a $Gal_1R$ antagonist, 10 nmol/kg, Tocris, #2697[30,31]) or M40 (a non-selective GalR antagonist, 3.3 nmol/kg, Tocris, #3425[32,33]) was injected bilaterally. After surgery, the pups were allowed to recover on a warming pad for ~5 min. Dams were briefly removed from their home cages before returning each pup to the nest. Pups were rubbed against their littermates to reduce the mother´s stress response. All mice underwent behavioral testing on P10. Subsequently, half of the mice were then transcardially perfused. Others were left to age to P45, when their locomotion was re-tested.

**Behavioral assays.** To test the spontaneous locomotor activity of infant mice, P10 pups were placed in perspex Petri dishes of 9 cm in diameter, with their activity video-recorded for off-line analysis at 24 °C for 8 min. Considering that infant mice are blind at this time, their behaviors along the walls of the miniature arenas were guided by their whiskers. Therefore, mice were tested both before and after whisker plucking to assess, if the removal of this sensory modality could reduce the pups' mobility or induces a change in the time spent in the center *vs.* border zone of the Petri dish (center: radius of 2.5 cm). Mice were returned to the nest between the recording sessions to limit stress and hypothermia. A total of $n = 11$ mice/group that had received scrambled *vs.* RNAi (both sexes), and $n = 8$ mice/group that were challenged pharmacologically were tested. Data were analyzed off-line using the Ethovision X15 software (Noldus). Cumulative distribution of movement was graphically illustrated as heat maps, and scaled from dark blue (immobility) to red (center of movement). On P45, adult mice were placed in PhenoTyper cage (28 cm in diameter, Noldus), with the spontaneous ambulatory activity recorded at 23 °C for 15 min[28]. Testing was performed in the dark. Data were analyzed on-line with Ethovision X15 software (Noldus).

### Statistics
Data were expressed as means ± sem. Data were analyzed using appropriate ANOVA designs. In histochemcial experiments, Tukey´s

*post-hoc* test for multiple comparisons was used with a reference age or genotype. Otherwise, data were evaluated using Student's *t*-test (two-tailed, unpaired). Prism 8 (GraphPad) was used for analysis, with $p < 0.05$ considered statistically significant.

## Reporting summary

Further information on research design is available in the Nature Portfolio Reporting Summary linked to this article.

## Data availability

Data generated in this study are made available for download in both raw and processed forms from the NCBI Gene Expression Omnibus, with accession number: GSE230180. An archived version of the submitted code together with data to reproduce figures from snRNA-seq data is available on Figshare with https://doi.org/10.6084/m9.figshare.22665352. Source data are provided in this paper.

## Code availability

The code developed and used in this report has been published at https://harkany-lab.github.io/Hevesi_2023/eda.html#Separate_analysis.

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

## Acknowledgements

We are indebted to D. Miloradovic and A. Tilscher for their administrative support. We also thank members of the Harkany laboratory for their discussion and appraisal of the original data. This work was supported by the Swedish Research Council (2023-03058, T.Ha; 2020-01688, T.Hö); Novo Nordisk Foundation (NNF23OC0084476, T.Ha.); Hjärnfonden (FO2022-0300, T.Ha.); the Arvid Carlsson Foundation (T.Hö.); the European Research Council (FOODFORLIFE, 2020-AdG-101021016; T.Ha.) and intramural funds of the Medical Neuroscience Cluster of the Medical University of Vienna (2021-1, T.Ha.). E.O.T. was supported by a scholarship from the Austrian Science Fund (FWF, DOC 33-B27). We thank Dr. E. Theodorsson (Linköping University, Linköping, Sweden) for his generous donation of the galanin antibody.

## Author contributions

T.Ha. and T.Hö. conceived the project. Z.H., T.Hö. and T.Ha. designed experiments. T.Ha. and T.Hö. procured funding. Z.H., J.B., E.O.T., C.A., A.R., S.B., M.C., T.K., S.L. and C.B. performed or managed experiments, and analyzed data. Z.H. and T.Ha. wrote the manuscript with input from all co-authors.

## Funding

## Competing interests

T.Hö. has stocks in Lundbeck A/S. All other authors declare that they have no conflict of interest.
