## [Peer Review File · Nature Communications]

Transient expression of the neuropeptide galanin modulates peripheral to central connectivity in the somatosensory thalamus during whisker development in miceREVIEWER COMMENTS

Reviewer #1 (Remarks to the Author):

Study Highlights

This short communication reports studies that demonstrate a novel role for a very highly-inducible neuropeptide, galanin, in a specific brainstem-thalamic circuit in postnatal brain. The authors identify that transient galanin expression in the ventrobasal thalamus coincides with whisker map development and drives related circuit wiring. The authors use temporal studies of the neuroanatomy and single-nucleus RNA-seq to identify complementary galanin and galanin receptor-1 expression between the thalamus and trigeminal afferents. They also use in vitro studies to assess the possible synthesis and release of galanin from thalamic neurons during a specific postnatal period and a genetic loss-of-function approach to reveal that galanin promotes glutamatergic synaptogenesis, based on evidence that the disruption of galanin/GalR1 signaling impairs whisker-dependent exploratory behaviors in young mice.

As the authors suggest, although there is existing evidence that galanin modifies circuit assembly and wiring (i.e., galanin acts to modulate axonal growth in vitro, and galanin has been shown to be a trophic factor for both peripheral and central neurons); nonetheless, the current studies do expand our knowledge of neuropeptide biology by highlighting that transient peptide expression in anatomical foci can specify neurocircuits, and that the impairment of this galanin signaling can manifest as altered behavior. This is a significant and important outcome.

More generally, the authors offer an arguably valid idea that this type of result identifies a novel avenue for future research to investigate how changes in the spatiotemporal distribution and levels of neuropeptides in the postnatal/juvenile brain could contribute to neurocircuit architecture, and whether alterations in these systems could be relevant to the pathobiology of neurodevelopmental neuropsychiatric disorders and their therapy.

Validity

The data are robust and in most cases main findings are validated using separate approaches, and anatomical, neurochemical and functional behavioral data reported. The interpretation and conclusions appear valid, and with my level of expertise in the various experimental fields, I could not identify any major flaws in the experimental approach or the data interpretation.

Significance

The results add support to earlier studies of the developmental role of galanin signaling and provide a quite direct demonstration of the importance of galanin/GalR1 signaling for brainstem-thalamus pathway development. In contrast to demonstrations of a similar involvement of other ligands and receptor systems in the definition of circuit architecture, these other systems are often less amenable to targeting in any therapeutic context, whereas G-protein-coupled receptors are the main targets of the majority of CNS therapies. Therefore these galanin/GalR1 findings do offer a better potential therapeutic opportunity than several other systems.

Data and Methodology

More formally, in my opinion, the methodology is sound and there are no flaws in the data analysis, and the interpretation and conclusions are sound. The research meets the expected high standards in the neuroscience field and there is sufficient detail provided in the methods for the research to be reproduced. Overall, the experimental approach appears solid and valid. The authors employed radioactive in situ hybridization and single-cell resolved in situ hybridization to identify galanin mRNA expression in the postnatal brain, in the ventrobasal thalamus, with a peak expression from postnatal day 4 to 10. They further confirm this transient expression during a potentially critical period using a Gal-Cre reporter mouse line, which identified thalamic neurons that were reporter+ in the thalamus at P7, but not in adulthood, confirming the transient activity of the galanin promoter in this area. These data established a novel site of galanin expression in the thalamus whose proposed function was a

physiological feature of neurocircuit assembly, associated with a short temporal window (P4-P10).

The authors also used an in vitro organotypic slice model to assess whether galanin release could be evoked from ventrobasal thalamic neurons during P4-P7. Organotypic slices prepared from P3, P6/7 and P20 thalami treated with KCl to release all transmitters produced increased galanin levels in the bath solution from P6/7 thalamic slices. The authors also reported supportive data that primary P0/P3 thalamic neurons plated and maintained for 4 days in vitro, displayed galanin expression in perikarya and along dendritic stretches; suggesting that, together, these data indicate that galanin can be released from the somatodendritic compartment of neurons populating the ventrobasal thalamus during the period of whisker circuit wiring.

As the whisker pathway matures during the first postnatal weeks when pups gain mobility, the authors explored the hypothesis that galanin, if released from neurons in the ventrobasal thalamus, would contribute to the maturation of incoming trigeminal axons. They used volumetric imaging of fluorescently-labeled neurons during the postnatal expansion of the ventrobasal thalamus to show their gradual increase in cell numbers until P21, providing further evidence of the transient nature of galanin expression. The authors also used single nucleus RNA-seq performed on the ventrobasal thalamus and Pr5 at P7 to demonstrate that the trigeminal Pr5 neurons expressed GalR1 (and not GalR2 or GalR3), providing the basis for these neurons to respond to galanin and use it to assist their axons to innervate thalamic relay neurons. This approach also identified that glutamate neurons, not GABA neurons, were involved in this pathway, and was followed by studies to map the glutamatergic inputs to the developing ventrobasal thalamus.

Lastly, a genetic loss-of-function study revealed that galanin is used for glutamatergic synaptogenesis, and that disruption of this process impaired whisker-dependent exploratory behaviors in young mice. The authors employed a handy 3D-printed stereotaxic apparatus customized for young mice to administer galanin siRNA into the ventrobasal thalamus of P5 animals, which produced an ~50% knock-down, to determine whether reduced galanin levels could reduce its chemotropic signal and alter the dynamics of synapse and circuit maturation in the thalamus, and thereby impair exploratory behaviors reliant on tactile stimuli. These studies revealed that galanin siRNA significantly reduced the density of glutamate neuron boutons apposing galanin thalamic neurons; and significantly reduced activity of P10 mice in a miniaturized open-field. Interestingly, in further testing of mice with their whiskers removed, a significant decrease in the motility and preference for exploring the center of the open-field arena was observed in sc-RNA-injected controls, whereas galanin siRNA-treated mice maintained their exploratory drive.

Overall, the data is of high-quality and extremely comprehensive, and the quality of presentation is excellent. The supplementary information provided with this short communication is also high-quality and the several figures are clear and well-presented and labelled.

Specific Comments

1. The authors should include the species (mouse) studied in the title.
2. The current study employs young mice of mixed sex, so the journal's policy on sex and gender considerations for studies involving vertebrate animals where relevant to the topic of study would appear to have been complied with. However, the authors should check to ensure that they have complied with the various recommendations particularly that the methods section includes whether sex and/or gender were considered in the study design, if relevant.
3. If there is no upper limit on the number of references that can be cited, the authors should endeavour to cite other references that support the role for galanin and GalR1 and GalR2 receptors in developing brain circuits, such as: Burazin TC et al. (2000) Galanin-R1 and -R2 receptor mRNA expression during the development of rat brain suggests differential subtype involvement in synaptic transmission and plasticity. *Eur J Neurosci* 12, 2901-2917. doi: 10.1046/j.1460-9568.2000.00184.x;

Komuro Y et al. (2021) The role of galanin in cerebellar granule cell migration in the early postnatal mouse during normal development and after injury. *J Neurosci.* 41:8725-8741. doi: 10.1523/JNEUROSCI.0900-15.2021; and Jungnickel SR et al. (2005) Induction of galanin receptor-1 (GalR1) expression in external granule cell layer of postnatal mouse cerebellum. *J Neurochem.* 92:1452-1462. doi: 10.1111/j.1471-4159.2004.02992; and any others.

4. During review of the manuscript, this reviewer has identified some terms and statements that could be clarified and improved. For example, on line 53, the authors should name the receptors, GalR1, GalR2 (if not R3). On line 58, the authors should provide more information and precision around the period "later". On line 63, "... a novel site of galanin expression in the thalamus, 'the function of which' could be associated with neurocircuit assembly ...". On line 66, ventrobasal thalamus 'represents' the primary

5. There are other terms/statements that might be modified for increased clarity, and these have been highlighted in an annotated pdf version of the article returned to the editorial office for consideration by the authors, when revising the article.

6. Formatting. The authors should use consistent spacing between numbers and units and consistent abbreviations or full terms (days, d, etc.) throughout.

Reviewer #2 (Remarks to the Author):

The authors elegantly show that the expression of the neuropeptide galanin in the ventrobasal thalamus is restricted to a short postnatal period and that GAL1R is expressed in trigeminal afferents. It is the first time that it has been shown that galanin seems to be important for glutamatergic synaptogenesis and whisker-dependent exploratory behavior in infant mice.

Still there are some important points to consider:

Regarding the expression of GAL1R, it would be important to determine whether galanin expressing cells are in close proximity to galanin.

Was galanin expression detected in Pr5 by RNAseq analysis?

The authors speculate that GAL1R expression could be a time locked feature for those neurons that actively undergo neuritogenesis at a given time. Could the authors perform some RNAscope experiments or mRNA expression analysis to verify this hypothesis?

Furthermore, the involvement of Gal1R could be further substantiated by silencing of GAL1R (similar to galanin) or the use of a Gal1R knock out animal.

Minor points

The authors should consider using the nomenclature of galanin receptors as recommended by the IPHAR/BPS guide of (see <https://www.guidetopharmacology.org/GRAC/FamilyDisplayForward?familyId=27>) that which would be galanin 1 receptor (GAL1 R),....

Abstract:

The authors state in the abstract that "Genetic loss of function revealed that galanin is used for glutamatergic synaptogenesis...". The reviewer does not think that a partial decrease by siRNA is a genetic loss of function. Thus, the authors should consider using the term "silencing of galanin expression" or omit the word genetic. It would certainly be an asset if the authors would involve

experiments with galanin knock out animals.

Methods:

Please place a space between a value and the unit. For example 0.1 M and not 0.1M

Most of the methods are provided in detail however, a more detailed description of the generation of the Gal-CreBAC::Ai14 mice (including the supplier of the stains used in the study) should be included in the methods section.

IHC: Can the authors explain why they have used different antibodies (see table 1) for the same proteins (RFP, GFP, VGLUT2)

qPCR: Similar to the antibodies, two different primer pairs are provided in table 3 for qPCR analysis. Which ones were used in which experiment?

Galanin ELISA: could the authors provide the amount of galanin detected for example per μg or mg protein in the supernatant rather than just reporting an OD.

Reviewer #3 (Remarks to the Author):

Hevesi et al. used radioactive riboprobe in situ hybridization to identify where in the brain the neuropeptide galanin is expressed. They identify it to be expressed in the neonatal ventrobasal thalamus during postnatal days (P) 4-10 but not during adulthood. Hevesi et al. show that galanin may act as guidance molecule of neurons regulating whisker development, and that knockdown of the galanin gene in the ventrobasal hypothalamus impairs mouse motility and whisker-dependent exploration. The manuscript is concisely written and based on an elegant set of experiments.

My review is focused on the single-cell parts of the manuscript. The authors used a single-nucleus protocol to generate their single-cell data (as opposed to a whole-cell approach). This is an important choice given that traditional whole-cell approaches applied to even early postnatal brain tissue typically yields suboptimal transcriptomics data especially for neurons. I am overall very enthusiastic about their work, and have the following comments which hopefully help to further increase the quality of their manuscript and the value of their single-cell data.

It would probably further improve the manuscript, if the authors could briefly summarize the quality of their single-cell data in the Results or Methods, and in a supplementary figure. E.g. state and show the the average numbers and distributions of unique genes and transcripts per cell across animals and timepoints.

The techniques used sequence the single-cell libraries (must to Illumina) should be stated in the Methods, as should the sequencing depth per cell.

It would be great if Hevesi et al. could briefly in the Methods describe how data were processed, quality controlled, filtered and normalized. Currently only a link to GitHub is provided. The should keep that link in the manuscript (the current GitHub repository link does not work yet, probably because the repository still is private), and add the bespoke brief description of the single-cell methods to the Methods.

To enable others to work with their valuable single-cell data, their time-resolved data should be made available through one of the publicly available database (e.g. through the Gene Expression Omnibus database).

Division of Molecular and Cellular
Neuroendocrinology
Department of Neuroscience

Tibor Harkany

Dr. Laura Mickelsen, *Associate Editor*
c/o **Dr. Fiona Carr**, *Senior Editor and Team Manager*
Nature Communications

RE: Point-by-point responses to Reviewer comments; NCOMMS-23-15769-T

Point-by-point responses to Reviewer #1:

Thank you for your appraisal of our manuscript.

Summary statement: ‘Study Highlights: This short communication reports studies that demonstrate a novel role for a very highly-inducible neuropeptide, galanin, in a specific brainstem-thalamic circuit in postnatal brain. The authors identify that transient galanin expression in the ventrobasal thalamus coincides with whisker map development and drives related circuit wiring. The authors use temporal studies of the neuroanatomy and single-nucleus RNA-seq to identify complementary galanin and galanin receptor-1 expression between the thalamus and trigeminal afferents. They also use in vitro studies to assess the possible synthesis and release of galanin from thalamic neurons during a specific postnatal period and a genetic loss-of-function approach to reveal that galanin promotes glutamatergic synaptogenesis, based on evidence that the disruption of galanin/GalR1 signaling impairs whisker-dependent exploratory behaviors in young mice.

As the authors suggest, although there is existing evidence that galanin modifies circuit assembly and wiring (i.e., galanin acts to modulate axonal growth in vitro, and galanin has been shown to be a trophic factor for both peripheral and central neurons); nonetheless, the current studies do expand our knowledge of neuropeptide biology by highlighting that transient peptide expression in anatomical foci can specify neurocircuits, and that the impairment of this galanin signaling can manifest as altered behavior. This is a significant and important outcome.

More generally, the authors offer an arguably valid idea that this type of result identifies a novel avenue for future research to investigate how changes in the spatiotemporal distribution and levels of neuropeptides in the postnatal/juvenile brain could contribute to neurocircuit architecture, and whether alterations in these systems could be relevant to the pathobiology of neurodevelopmental neuropsychiatric disorders and their therapy.

Validity: The data are robust and in most cases main findings are validated using separate approaches, and anatomical, neurochemical and functional behavioral data reported. The interpretation and conclusions appear valid, and with my level of expertise in the various experimental fields, I could not identify any major flaws in the experimental approach or the data interpretation.

Significance: The results add support to earlier studies of the developmental role of galanin signaling and provide a quite direct demonstration of the importance of galanin/GalR1 signaling for brainstem-thalamus pathway development. In contrast to demonstrations of a similar involvement of other ligands and receptor systems in the definition of circuit architecture, these other systems are often less amendable to targeting in any therapeutic context, whereas G-protein-coupled receptors are the main targets of the majority of CNS therapies. Therefore these galanin/GalR1 findings do offer a better potential therapeutic opportunity than several other systems.

Postal address
Solnavägen 9
S-17165 Solna
Sweden

Org. number 202100 2973

Visiting address
Biomedicum 7D
Solnavägen 9
S-17165 Solna
Sweden

Telephone
+46 8 524 87070, direct

E-Mail
Tibor.Harkany@ki.se
Web
ki.se

Data and Methodology: More formally, in my opinion, the methodology is sound and there are no flaws in the data analysis, and the interpretation and conclusions are sound. The research meets the expected high standards in the neuroscience field and there is sufficient detail provided in the methods for the research to be reproduced. Overall, the experimental approach appears solid and valid. The authors employed radioactive in situ hybridization and single-cell resolved in situ hybridization to identify galanin mRNA expression in the postnatal brain, in the ventrobasal thalamus, with a peak expression from postnatal day 4 to 10. They further confirm this transient expression during a potentially critical period using a Gal-Cre reporter mouse line, which identified thalamic neurons that were reporter+ in the thalamus at P7, but not in adulthood, confirming the transient activity of the galanin promoter in this area. These data established a novel site of galanin expression in the thalamus whose proposed function was a physiological feature of neurocircuit assembly, associated with a short temporal window (P4-P10).

The authors also used an in vitro organotypic slice model to assess whether galanin release could be evoked from ventrobasal thalamic neurons during P4-P7. Organotypic slices prepared from P3, P6/7 and P20 thalami treated with KCl to release all transmitters produced increased galanin levels in the bath solution from P6/7 thalamic slices. The authors also reported supportive data that primary P0/P3 thalamic neurons plated and maintained for 4 days in vitro, displayed galanin expression in perikarya and along dendritic stretches; suggesting that, together, these data indicate that galanin can be released from the somatodendritic compartment of neurons populating the ventrobasal thalamus during the period of whisker circuit wiring.

As the whisker pathway matures during the first postnatal weeks when pups gain mobility, the authors explored the hypothesis that galanin, if released from neurons in the ventrobasal thalamus, would contribute to the maturation of incoming trigeminal axons. They used volumetric imaging of fluorescently-labeled neurons during the postnatal expansion of the ventrobasal thalamus to show their gradual increase in cell numbers until P21, providing further evidence of the transient nature of galanin expression. The authors also used single nucleus RNA-seq performed on the ventrobasal thalamus and Pr5 at P7 to demonstrate that the trigeminal Pr5 neurons expressed GalR1 (and not GalR2 or GalR3), providing the basis for these neurons to respond to galanin and use it to assist their axons to innervate thalamic relay neurons. This approach also identified that glutamate neurons, not GABA neurons, were involved in this pathway, and was followed by studies to map the glutamatergic inputs to the developing ventrobasal thalamus.

Lastly, a genetic loss-of-function study revealed that galanin is used for glutamatergic synaptogenesis, and that disruption of this process impaired whisker-dependent exploratory behaviors in young mice. The authors employed a handy 3D-printed stereotaxic apparatus customized for young mice to administer galanin siRNA into the ventrobasal thalamus of P5 animals, which produced an ~50% knock-down, to determine whether reduced galanin levels could reduce its chemotropic signal and alter the dynamics of synapse and circuit maturation in the thalamus, and thereby impair exploratory behaviors reliant on tactile stimuli. These studies revealed that galanin siRNA significantly reduced the density of glutamate neuron boutons apposing galanin thalamic neurons; and significantly reduced activity of P10 mice in a miniaturized open-field. Interestingly, in further testing of mice with their whiskers removed, a significant decrease in the motility and preference for exploring the center of the open-field arena was observed in sc-RNA-injected controls, whereas galanin siRNA-treated mice maintained their exploratory drive.

Overall, the data is of high-quality and extremely comprehensive, and the quality of presentation is excellent. The supplementary information provided with this short communication is also high-quality and the several figures are clear and well-presented and labelled.'

We were glad to learn your positive view on our study, particularly that you have found its outcome '*significant and important*'. It was most reassuring to read that you judged our work of '*high-quality and extremely comprehensive, and the quality of presentation is excellent*'. At the same time, we thank you for your specific comments, which we have addressed both in the revised manuscript and in this letter. Accordingly, please find our specific replies to your queries below.

Q1: 'The authors should include the species (mouse) studied in the title.'

The title was amended as requested.

Q2: 'The current study employs young mice of mixed sex, so the journal's policy on sex and gender considerations for studies involving vertebrate animals where relevant to the topic of study would appear to have been complied with. However, the authors should check to ensure that they have complied with the various recommendations particularly that the methods section includes whether sex and/or gender were considered in the study design, if relevant.'

The 'Methods' section has been amended. Moreover, **we have performed separate statistics on both female and male offspring where possible**, even if the smaller cohort sizes limited statistical power. These data were added to the 'Source Data File'. **Sex-specific data** points were also labelled in the relevant figure panel by distinct color.

Q3: 'If there is no upper limit on the number of references that can be cited, the authors should endeavour to cite other references that support the role for galanin and GalR1 and GalR2 receptors in developing brain circuits, such as: Burazin TC et al. (2000) Galanin-R1 and -R2 receptor mRNA expression during the development of rat brain suggests differential subtype involvement in synaptic transmission and plasticity. *Eur J Neurosci* 12, 2901-2917. doi: 10.1046/j.1460-9568.2000.00184.x; Komuro Y et al. (2021) The role of galanin in cerebellar granule cell migration in the early postnatal mouse during normal development and after injury. *J Neurosci*. 41:8725-8741. doi: 10.1523/JNEUROSCI.0900-15.2021; and Jungnickel SR et al. (2005) Induction of galanin receptor-1 (GalR1) expression in external granule cell layer of postnatal mouse cerebellum. *J Neurochem*. 92:1452-1462. doi: 10.1111/j.1471-4159.2004.02992; and any others.'

Thank you for asking this. Indeed, the original version of our submission complied with the restricted format of a 'Brief communication' in *Nature Neuroscience*. As per your request, **we have expanded the introduction and discussion of our report, included the above references**, and more.

Q4-Q6 'During review of the manuscript, this reviewer has identified some terms and statements that could be clarified and improved. For example, on line 53, the authors should name the receptors, GalR1, GalR2 (if not R3). On line 58, the authors should provide more information and precision around the period "later". On line 63, "... a novel site of galanin expression in the thalamus, 'the function of which' could be associated with neurocircuit assembly ...". On line 66, ventrobasal thalamus 'represents' the primary'; 'There are other terms/statements that might be modified for increased clarity, and these have been highlighted in an annotated pdf version of the article returned to the editorial office for consideration by the authors, when revising the article'; 'Formatting. The authors should use consistent spacing between numbers and units and consistent abbreviations or full terms (days, d, etc.) throughout.'

Thank you for sending us the annotated manuscript. We have **carefully edited this revision** to remove any ambiguous terminology, provided succinct anatomical and functional **definitions**, and standardized **formatting** throughout.

Point-by-point responses to Reviewer #2:

We thank the Reviewer for their appraisal of our report.

Summary statement: 'The authors elegantly show that the expression of the neuropeptide galanin in the ventrobasal thalamus is restricted to a short postnatal period and that GAL1R is expressed in trigeminal afferents. It is the first time that it has been shown that galanin seems to be important for glutamatergic synaptogenesis and whisker-dependent exploratory behavior in infant mice. Still there are some important points to consider: ...'

We thank the Reviewer for their appreciation of our study, considering it '*elegant*' and one that shows that '*galanin seems to be important for glutamatergic synaptogenesis and whisker-dependent exploratory behavior in infant mice*'. We also appreciate the specific queries, which have helped us to further strengthen our conclusions. Our replies to your individual points are as follows.

Q1: 'Regarding the expression of GAL1R, it would be important to determine whether galanin expressing cells are in close proximity to galanin.'

We presume your question was meant to ask if GAL1R-containing axons appose galanin⁺ somata and dendrites. Given the lack of a GAL1R-specific antibody, it is methodologically nigh impossible (without transgene technologies) to address this question because it would require a combined *in situ* hybridization/immunohistochemical study at single synapse resolution with an anti-GAL1R antibody to label axon terminals. We have also inquired if Dr. Wynick had Galr1-mCherry (GFP) knock-in mice (C-terminal tagging) available. Unfortunately, he has closed his laboratory and henceforth mice can no longer be shared. Nevertheless, in his earlier publication (Ref: Kerr N, Holmes FE, Hobson SA, Vanderplank P, Leard A, Balthasar N, Wynick D. The generation of knock-in mice expressing fluorescently tagged galanin receptors 1 and 2. *Mol Cell Neurosci.* 68: 258-271 (2015)) one can find: '*In brain, high levels of Gal1-mCherry immunofluorescence were detected within thalamus, hypothalamus and amygdala, with a high density of nerve endings...*'. Even though this report only showed images of adult mice only, and their resolution seemed limited, they give confidence that GAL1R might be enriched in the thalamus with some labeling possibly localized to axon terminals. Even more so, Fig. 6J,M of this reference showed GAL1R-mCherry transported along axons, providing morphological support to our mechanistic data. We would, nevertheless, caution that mCherry was inserted in the C-terminus of *Galr1*, which could interfere with signal transduction downstream from chimeric receptors (noting that G protein recruitment is typically at the C-terminus of GPCRs), potentially affecting functional neurocircuit analysis.

As a best possible alternative, and in view of your next question, we have two separate sets of data that might offer some insights. In **revised Fig. 3**, we show that *Gal* and *Galr1* expression patterns are mutually exclusive in the ventrobasal thalamus (VB) and Pr5. Galanin expression was abundant in the VB. In **revised Fig. 4**, we demonstrated that VGLUT2⁺ axon terminals encircle galanin⁺ perikarya in the VB, with their density changing upon RNAi-mediated silencing of galanin expression. Thus, and even if indirectly, these two sets of data suggest that *i*) galanin-GAL1R signaling is unlikely to be cell autonomous (they are not co-expressed in Pr5), and instead *ii*) galanin released from VB neurons could affect the growth and targeting of afferent axons. Moreover, and even if somewhat circumstantial, we have over-expressed GAL1Rs (GFP-tagged) in PC12 cells (**revised Extended Data Figure 2d**) and could observe their presence in filopodia/lamellipodia. These receptors were internalized and trafficked to the soma upon M617 superfusion.

In view of the above, **we have discussed this point, expanded Extended Data Fig. 2, and inserted a statement on potential limitations and the use of genetic tags in combination with histochemistry.**

Q2: 'Was galanin expression detected in Pr5 by RNAseq analysis?'

We have **revised Fig. 3** to show that single-nucleus RNA-seq **did not find galanin expression in Pr5.**

Q3: 'The authors speculate that GAL1R expression could be a time locked feature for those neurons that actively undergo neuritogenesis at a given time. Could the authors perform some RNAscope experiments or mRNA expression analysis to verify this hypothesis?'

Thank you for asking this. We used single-nucleus RNA-seq data to address this question. We subset *Galr1*⁺ neurons of the Pr5 and searched for genes that are associated with neuritogenesis (cytoskeletal dynamics and expansion), and growth cone steering decisions. We have **inserted the data numerically in the 'Results' section** for the sake of simplicity.

Q4: 'Furthermore, the involvement of Gal1R could be further substantiated by silencing of GAL1R (similar to galanin) or the use of a Gal1R knock out animal.'

Indeed, these are relevant experiments. However, gene silencing was deemed as often having off-target effects (as noted by the other Peer-reviewers), and injecting the Pr5 in infant mice is an extremely complicated, if not impossible, task. For *Galr1*, knock-out mice exist. However, to the best of our knowledge, there is no commercially available conditional allele (floxed) for the receptor. As customary, a constitutive knock-out line would unlikely be accepted for a developmental biology study. Therefore, we have taken an alternative **pharmacological approach**: either a GAL1R-selective agonist (M617) or a commercially available, non-selective GALR antagonist (M40) was injected in a paradigm reminiscent of the RNAi experiment. **Data were shown in revised Fig. 4 and Supplementary Fig. 4,5.** M617 effects opposed those of the RNAi approach for both neuroanatomy and behavior. M40 mostly produced effects reduced vs. control, albeit lacking statistical significance (likely due to receptor internalization and degradation). Together, **our RNAi and M617 data present loss-of-function and gain-of-function phenotypes, respectively**, through selectively modulating either galanin expression or GAL1R activity.

Q5 (from here on minor points): 'The authors should consider using the nomenclature of galanin receptors as recommended by the IPHAR/BPS guide of (see <https://www.guidetopharmacology.org/GRAC/FamilyDisplayForward?familyId=27>) that which would be galanin 1 receptor (GAL1 R),....'

We appreciate your noting this. We have **edited the manuscript** and referred to protein as **Gal₁R** (as per the guide you have referred to), and the gene as ***Galr1*** throughout.

Q6: 'The authors state in the abstract that "Genetic loss of function revealed that galanin is used for glutamatergic synaptogenesis...". The reviewer does not think that a partial decrease by siRNA is a genetic loss of function. Thus, the authors should consider using the term "silencing of galanin expression" or omit the word genetic. It would certainly be an asset if the authors would involve experiments with galanin knock out animals.'

We have edited this part of the text as requested.

Q7: 'Please place a space between a value and the unit. For example 0.1 M and not 0.1M'

The **manuscript was proofread** to standardize these expressions and introduce the requested changes.

Q8: 'Most of the methods are provided in detail however, a more detailed description of the generation of the Gal-CreBAC::Ai14 mice (including the supplier of the stains used in the study) should be included in the methods section.'

The **'Methods' section was expanded** to improve clarity.

Q9: 'IHC: Can the authors explain why they have used different antibodies (see table 1) for the same proteins (RFP, GFP, VGLUT2)'

Some of the **experiments were performed (and reproduced) independently at two sites** (Stockholm, Vienna). When doing so, we have placed emphasis on quality controls and for galanin and VGLUT2 we have purposefully used different antisera. Moreover, please note that some of the amplification methods were also different (e.g., TSA amplification vs. conventional indirect detection). Thus, we are confident that key findings of this report are correct. For GFP, an error was unfortunately left in the antibody list since the Abcam #6662 antibody is a FITC-conjugated anti-GFP antibody of goat origin. This has been **corrected**.

Q10: 'qPCR: Similar to the antibodies, two different primer pairs are provided in table 3 for qPCR analysis. Which ones were used in which experiment?'

Thank you for drawing our attention to this point. We have **removed the primer pair** (preprogalanin) that is irrelevant to the present study.

Q11: 'Galanin ELISA: could the authors provide the amount of galanin detected for example per μg or mg protein in the supernatant rather than just reporting an OD.'

Thank you for asking this. Data were expressed as **pg/ml supernatant with Fig. 2e,e₁ updated accordingly**.

Point-by-point responses to Reviewer #3:

Thank you for your expert opinion on our manuscript.

Summary statement: 'Hevesi et al. used radioactive riboprobe in situ hybridization to identify where in the brain the neuropeptide galanin is expressed. They identify it to be expressed in the neonatal ventrobasal thalamus during postnatal days (P) 4-10 but not during adulthood. Hevesi et al. show that galanin may act as guidance molecule of neurons regulating whisker development, and that knockdown of the galanin gene in the ventrobasal hypothalamus impairs mouse motility and whisker-dependent exploration. The manuscript is concisely written and based on an elegant set of experiments.'

My review is focused on the single-cell parts of the manuscript. The authors used a single-nucleus protocol to generate their single-cell data (as opposed to a whole-cell approach). This is an important choice given that traditional whole-cell approaches applied to even early postnatal brain tissue typically yields suboptimal transcriptomics data especially for neurons. I am overall very enthusiastic about their work, and have the following comments which hopefully help to further increase the quality of their manuscript and the value of their single-cell data.'

We were glad you found the paper '*concisely written and based on an elegant set of experiments*'. We particularly appreciated your appraisal of the single-nucleus RNA-seq data that form an integral part of the manuscript and were encouraged by your enthusiasm. Our responses to your specific comments are as follows.

Q1: 'It would probably further improve the manuscript, if the authors could briefly summarize the quality of their single-cell data in the Results or Methods, and in a supplementary figure. E.g. state and show the the average numbers and distributions of unique genes and transcripts per cell across animals and timepoints.'

We have **significantly extended the 'Methods' section**, described *i)* **data pre-processing**, *ii)* **quality controls including droplet selection, ambient RNA removal, doublet detection**, *iii)* **further filtering**, *iv)* **clustering**, and *v)* **choosing marker genes**. We would hope this description conforms to the standards in this field fully.

Q2: 'The techniques used sequence the single-cell libraries (must to Illumina) should be stated in the Methods, as should the sequencing depth per cell.'

Indeed, we used Illumina sequencing (**Illumina HiSeq 4000**). **The median nCount was 5,142 and 8,434 for the Pr5 and VB nuclei, respectively**. Please see: https://harkany-lab.github.io/Hevesi_2023/eda.html for details.

Q3: 'It would be great if Hevesi et al. could briefly in the Methods describe how data were processed, quality controlled, filtered and normalized. Currently only a link to GitHub is provided. The should keep that link in the manuscript (the current GitHub repository link does not work yet, probably because the repository still is private), and add the bespoke brief description of the single-cell methods to the Methods.'

Please see our answers to **Q1 for details**. Unfortunately, the **GitHub link was broken** in the original manuscript, and this is why it was inaccessible (https://harkany-lab.github.io/Hevesi_2023/eda.html). This has been corrected, with its publication date (together with the GEO/GSE) postponed until April 1st, 2024.

Q4: 'To enable others to work with their valuable single-cell data, their time-resolved data should be made available through one of the publicly available database (e.g. through the Gene Expression Omnibus database).'

Thank you for asking this. We have amended the manuscript with a '**Data availability statement**' and submitted the raw data to GEO with accession number **GSE230180**.

REVIEWERS' COMMENTS

Reviewer #1 (Remarks to the Author):

In this revised manuscript, the authors have provided a substantially revised and improved article containing additional data that is supported by detailed and satisfactory responses to each of the many specific points raised by the expert reviewers.

As a result, this well written article contains substantial complementary data of significance for the field that supports the major conclusions of the authors.

Furthermore, the methodology employed has been further expanded and explained in the methods section, which would enable the work to be reproduced or extended by other investigators; and some of the gene expression data has been lodged in a database for others to explore as required. I congratulate the authors on their efforts.

Reviewer #2 (Remarks to the Author):

The authors have addressed all my points satisfactorily.

Reviewer #3 (Remarks to the Author):

Thank you for addressing all my comments and providing the R markdown script, which will undoubtedly benefit others interested in analyzing their or similar data. Congratulations once again on your commendable paper.

Division of Molecular and Cellular
Neuroendocrinology
Department of Neuroscience

Tibor Harkany

Dr. Laura Mickelsen, *Associate Editor*
Nature Communications

RE: Point-by-point responses to Reviewer comments; NCOMMS-23-15769A

Point-by-point responses to Reviewer #1-#3:

We wholeheartedly thank the three Expert Referees for scrutinizing our manuscript. We were glad to learn that none of them had any further question or query, and found our manuscript '*substantially revised and improved*' that, with their help, became a '*commendable paper*'.